# Research on grain production efficiency in China's main grain producing areas from the perspective of financial support

Dehua Zhang◉, Haiqing Wang◉, Sha Lou◉*, Shen Zhong◉

Institute of Finance, Harbin University of Commerce, Harbin, China

◉ These authors contributed equally to this work.
* 102896@hrbcu.edu.cn

**Data Availability Statement:** All relevant data are within the manuscript and its Supporting Information files.

## Abstract

Grain production is vital to the national economy and people's livelihood, and improving grain production efficiency is of great significance to the sustainable development of China's economy and society. From the perspective of financial support, using the DEA global Malmquist productivity index model and based on the data of 13 main grain producing areas in China from 2001 to 2017, this paper discusses the evolution characteristics and regional distribution differences of the total factor productivity index of grain production in China's main grain producing areas. The results show that from 2001 to 2017, the total factor productivity index of grain production in China's main grain producing areas showed an overall fluctuation trend of gradual decline, with an average annual decline of 7.3%. From the perspective of spatial analysis, the grain production efficiency in China's main grain producing areas is characterized by uneven spatial development, which is generally manifested as the decreasing trend from the central region to the eastern and western regions. Meanwhile, it can be seen from the decomposition index that the change of total factor productivity of grain production in China's main grain producing areas mainly depends on the change of technical efficiency.

## Introduction

According to the 2018 China Grain Development Report released by the China Food and Strategic Reserves Administration, China's total grain output in 2017 was 617.91 million tons, with the per capita grain production being 472 kilograms. Meanwhile, China's per capita grain consumption is 479 kilograms, making the self-sufficiency rate of grain only 82.3%, which is lower than the world security standard of 90%. The contradiction between present grain supply and future demand remains a core issue that China needs to resolve urgently. Judging from the actual situation of China's grain production, the supply of cultivated land has reached its limit [1]. China has always adhered to the red line of 1.8 billion mu of farmland. In the foreseeable future, arable land will be under further pressure, and the overall food supply will be severely tested [2]. In this context, it is particularly important to improve grain production efficiency in

**Funding:** This work was supported by Heilongjiang Province Philosophy and Social Science Fund Project (18JYE669); Youth Innovative Talents Training Program in General Undergraduate Colleges and Universities in Heilongjiang Province (UNPYSCT2018125); Special support project for postdoctoral in Heilongjiang Province (LBHZ19073); Special plan for top young talents of Harbin University of Commerce (2020CX42); Graduate Innovation Project of Harbin University of Commerce (YJSCX2020-643HSD).

**Competing interests:** The authors have declared that no competing interests exist.

the main grain producing areas in China [3]. Therefore, the study on the measurement of grain production efficiency has become a very important issue. Through the measurement of grain production efficiency, we can explore the effective way to improve the grain production efficiency, and put forward reasonable solutions [4, 5]. At present, the research on grain production efficiency mainly concentrates on DEA model [6, 7], scholars usually select land resources, labor resources and other factors of production as input indicators [8, 9]. However, at the 19th National Congress of the Communist Party of China (CPC), China stressed the importance of increasing financial support for agricultural production, and continued to implement the fiscal policy of supporting agriculture under the background of the strategy of "balancing urban and rural development" [10]. China will establish a long-term mechanism for ensuring steady growth of funding for agricultural production, and continue to increase funding for agricultural infrastructure construction and comprehensive agricultural development [11]. China put forward and implemented the Rural Revitalization Strategy, promoted innovation in rural financial institutions, and ensured the effective and precise supply of funds for agricultural production [12]. It can be seen that food subsidies can also be converted into factors of production [13]. The grain production efficiency can be measured more accurately if the factor of grain subsidy is used to study the grain production efficiency [14]. Therefore, the innovation point of this paper is to include agricultural loan into input index from the perspective of financial support, by discussing the evolution characteristics and regional distribution differences of TFP index of grain production in China's main grain producing areas, it is of great significance to improve the grain production efficiency and social harmony and stability in China's main grain producing areas [15].

The chapter structure of this paper is as follows: the second chapter introduces the literature of grain production efficiency; the third chapter introduces the model setting, index selection and sample selection, the fourth chapter is the empirical analysis and explanation of grain production efficiency; conclusions and related policy recommendations are explained in the fifth chapter.

## Literature review

At present, the research on grain production efficiency mainly concentrates on the following two aspects:

First is the calculation method of grain production efficiency. James Odeck (2007) used the combination method of parametric SFA and non-parametric DEA to measure the rice production capacity and technical efficiency of 19 professional grain farm producers in eastern Norway based on the sample data from 1987 to 1997. Studies have shown that technological efficiency is an important factor in improving food production capacity [16]. Alene compared the traditional methods and modern analytical techniques for the measurement of grain production efficiency based on the data of single indicator grain planting area of grain production in two climatic regions of Ethiopia, found that modern analytical techniques had advantages in the measurement of grain production efficiency [17]. Zhang Qinan et al (2018) used three-stage DEA model and Tobit regression model to analyze the influencing factors of China's grain production efficiency that based on the provincial panel data of main grain producing areas from 2006 to 2016. The results show that the effective irrigation area and the amount of chemical fertilizer have a positive effect on the grain production efficiency, but the effect is not obvious. Maintaining a certain grain retail price index is conducive to improving the grain production efficiency, while the improvement of the total power of agricultural machinery and the level of regional economic development will hinder it [18]. Linn Maenhout et al (2019) analyzed the profitability and efficiency of rice production in the Irrawaddy region of

Myanmar using descriptive statistics and data envelope analysis (DEA) based on the original data of 130 randomly sampled farmers. It was found that the rice production efficiency could be improved by establishing farmer cooperatives to expand the operation scale and improving the scale efficiency. In addition, governments should intervene to reduce input prices, control the quality of input seeds, and establish appropriate financial crop insurance mechanisms [19]. It can be found that the measurement methods of grain production efficiency in previous literatures are mainly based on Stochastic Frontier Analysis (SFA) and Data Envelopment Analysis (DEA).

The second is the index and data selection for measuring the efficiency of grain production. Raghbendra Jha et al. (2000) used data envelopment analysis (DEA) to measure the allocation efficiency and technical efficiency of wheat production in the Indian state of Punjab, based on data from 300 farms spanning 1981–1983. Their results showed that both types of efficiency are closely related to a farm's scale and output. Larger firms are more efficient, produce more output, and thus, deliver greater benefits to farmers [20]. Yuna Alemdar et al. (2006) used the DEA model to calculate the technical efficiency of farms based on wheat-planting data from southeastern Anatolia in Turkey from 2000–2001. The results showed that land fragmentation is an important reason for the low efficiency of food production [21]. Amos K et al. (2010) considered the composite index of national food production in Ghana, empirical data on the total agricultural population and its growth rate, the total area of national agricultural land, and the planting area dedicated to each main food crop from 1962 to 2004, and concluded that food production was positively correlated with planting area and population [22]. Yao (2011) analyzed the effect of input factors such as labor, fertilizer, and capital on crop production, measured based on the area of cultivated land, applying the stochastic frontier production function to panel data spanning 1987–1992 from 30 Chinese provinces. The authors concluded that there are significant province-level technical inefficiencies and regional differences [23]. These literature reviews reveal that in previous studies, the study period is often short, the year is not new, and the input–output index of grain production efficiency is mostly based on labor, land, and capital. Few scholars consider the perspective of financial support.

Therefore, compared to previous studies, the innovation of this paper is mainly in the following three aspects: This study, thus, makes the following contributions to the literature. First, in terms of data selection, this study is based on grain production input–output data, the new innovation is to take agricultural loan as input index. Second, the strength of DEA lies in the fact that it is a nonparametric method, and there is no need to make many prior assumptions. It allows for multiple outputs and inputs but does not allow the separation of random error from inefficacy [24]. Considering scale heterogeneity and regional heterogeneity, this study develops a global Malmquist index model based on the classic DEA. Third, in comparison to the previous literature, we use recent data with longer time dimension, which is more in line with the actual situation, allowing us to evaluate grain production efficiency more accurately. In doing so, the study potentially provides a comprehensive explanation for the spatial–temporal spillover effect of grain production efficiency and a more reliable theoretical basis to aid government decision-making.

## Data and methods

### DEA global Malmquist model

Assuming that the *k*th decision making unit (DMU) has *F* inputs, the input variable is denoted as $X_{ik}$, written as a vector $X = (X_{1k}, X_{2k}, \ldots, X_{fk})^Q$; there are *g* outputs. The output variable is denoted as the vector $y = (y_{1k}, y_{2k}, \ldots, y_{gk})^Q$. Further, $Q = \{(x,y)|put\ into\ x\ product\ y\}$ refers

to all possible production sets that may be formed in the production process, where $Q^q$ is the collection of inputs and outputs at all $q$.

Referring to Pastor and Lovell (2005) [25], when returns to scale are constant, production technology is defined as current production technology $Q_c^q$ and global production technology $Q_c^G$ The current production technology is:

$$Q_c^q = \left\{ (x^q, y^q) \middle| put\ into\ x^q product\ y^q \right\} \tag{1}$$

The global production technology is

$$Q_c^G = Q_c^1 \cup \cdots \cup Q_c^Q \tag{2}$$

The current production technology $Q_c^q$ and the global production technology $Q_c^G$ subscript "C" both satisfy the assumption of constant returns to scale. The TFP index is defined from the perspective of production technology [26]:

$$M_c^p(x^q, y^q, x^{q+1}, y^{q+1}) = \frac{D_c^p(x^{q+1}, y^{q+1})}{D_c^p(x^q, y^q)} \tag{3}$$

$$D_c^p(x^q, y^q) = \min\{\vartheta > 0 | (x, y/\vartheta) \in Q_c^p\}(p = q, q+1) \tag{4}$$

The above formula $D_c^p(x^{q+1}, y^{q+1})$ is the output-oriented distance function.

Because $M_c^q(x^q, y^q, x^{q+1}, y^{q+1}) \neq M_c^{q+1}(x^q, y^q, x^{q+1}, y^{q+1})$, the academia uses the geometric mean of the product of these two indices to define the TFP of the Malmquist index for the current period, which can avoid the arbitrariness of period selection:

$$M_c(x^q, y^q, x^{q+1}, y^{q+1}) = [M_c^q(x^q, y^q, x^{q+1}, y^{q+1}) * M_c^{q+1}(x^q, y^q, x^{q+1}, y^{q+1})]^{1/2} \tag{5}$$

The TFP index is defined from the production technology $Q_c^G$:

$$M_c^G(x^q, y^q, x^{q+1}, y^{q+1}) = \frac{D_c^G(x^{q+1}, y^{q+1})}{D_c^G(x^q, y^q)} \tag{6}$$

The yield distance function at this point: $D_c^G(x, y) = \min\{\vartheta > 0 | (x, y/\vartheta) \in Q_c^G\}$.

The current Malmquist productivity index and the global Malmquist productivity index have one thing in common [27]. That is $(x^{q+1}, y^{q+1})$ and $(x^q, y^q)$ the difference is that the two indexes are compared with different benchmarks. The technical front of the global Malmquist productivity index is established using all the observed values, which is a good way to avoid the problem of randomness of period selection. $M_c^G$ can be broken down into the following situations:

$$M_c^G(x^q, y^q, x^{q+1}, y^{q+1}) = \frac{D_c^{q+1}(x^{q+1}, y^{q+1})}{D_c^q(x^q, y^q)} \times$$

$$\left\{ \frac{D_c^G(x^{q+1}, y^{q+1})}{D_c^{q+1}(x^{q+1}, y^{q+1})} \times \frac{D_c^q(x^q, y^q)}{D_c^G(x^q, y^q)} \right\} = EC \times TECH \tag{7}$$

Among them, EC is technical efficiency change, tech is technical change, $M_c^G(x^q, y^q, x^{q+1}, y^{q+1})$ is usually used to measure TFP and is often recorded as TFPCH. The definition of total factor productivity is the comprehensive productivity of production unit as each factor in the system and the portion of output not explained by the amount of inputs used in production. In addition. According to Fare (1994) [28], EC can be further decomposed into pure technical change (BPC) and scale efficiency change (SECH), so the final

decomposition formula is:

$$TFPCH = EC \times TECH = BPC \times SECH \times TECH \qquad (8)$$

That is, the TFP index is decomposed into technical efficiency changes (EC) and technological progress changes (TECH), and technical efficiency changes can be further decomposed into pure technical efficiency changes (BPC) and scale efficiency changes (SECH). In the output results of maxdea software, whether it is input-oriented or output oriented, the meaning of Malmquist index is that greater than 1 means higher productivity and less than 1 means lower productivity [29].

## Index selection, sample selection and data sources

The grain production efficiency is the reflection of the comprehensive utilization degree of factor input and the proportion of input and output in the process of regional grain production. It is the most commonly used manifestation of the allocation between regional cultivated land resources, labor force, financial support and the economic benefits generated. Based on the predecessors' research results and combining with the actual situation of main grain producing areas. Following the principles of availability and representativeness of index selection, four input indexes and one output index were selected to construct the evaluation system of grain production efficiency. The indexes are explained as follows. Input index:

(1) Agricultural loans. It is the general term for loans granted by Agricultural Bank of China and other rural financial institutions to funds needed for agricultural production [30].

(2) Agricultural personnel. Rural population of more than 16 years of age to participate in agricultural production and business activities and obtain physical or monetary income [31].

(3) Grain planting area. The actual area of land sown or transplanted with crops, which can be used for regular tillage [32].

(4) Effective irrigation area. The effective irrigation area is the farmland area with flat plot, certain water source and irrigation facilities, which can be irrigated normally in general years. It is the sum of irrigated area in paddy field and dry land for normal irrigation. (New input indicator added according to reviewers' opinions.)

Food security is an important strategic goal of national food production and grain output is the best indicator to measure national food security. Output index:

(5) Grain output. Refers to the total amount of grain produced within the calendar year of agricultural producers and operators [33].

To sum up, this paper uses four indicators of agricultural loans, agricultural employees, effective irrigation area and grain planting area as input indicators, and grain output as output indicators. The index data published by the China Bureau of Statistics and those in the statistical yearbooks are not unified; only the provincial data are unified, while farm-level data do not have a unified branch for investigation. In this paper, the index data of grain production efficiency in China's main grain producing areas are obtained from 'China Statistical Yearbook', 'China Financial Yearbook', 'China Rural Statistical Yearbook' and the website database of the National Bureau of Statistics. Descriptive statistics for the indicators are presented in Table 1.

The samples of this paper are 13 main grain producing areas in China. The main grain producing areas refer to the key grain production areas [34], where the local climatic conditions, environment and soil are suitable for crop growth, and have certain scale and economic benefits. This paper divides China's 13 main grain producing provinces into three regions: the

**Table 1. Descriptive statistics of input and output indexes.**

| Criterion layer | Index layer | Unit | Max | Min | Mean | Std. Dev. | Obs. |
|---|---|---|---|---|---|---|---|
| Input indexes | Agricultural personnel | Ten thousand people | 4915 | 620 | 2129 | 1305 | 221 |
| | Grain planting area | Thousands of hectares | 14283 | 2564 | 6558 | 2404 | 221 |
| | Agricultural loans | One hundred million yuan | 31080 | 87 | 4696 | 6240 | 221 |
| | Effective irrigation area | Thousands of hectares | 6056 | 1294 | 3240 | 1256 | 221 |
| Output index | Grain output | Ten thousand tons | 7616 | 1239 | 3206 | 1271 | 221 |

eastern region (Jiangsu, Liaoning, Hebei, Shandong), the central region (Heilongjiang, Jilin, Henan, Hubei, Hunan, Anhui, Jiangxi, Inner Mongolia) and the western region (Sichuan Province) [35]. The differences of TFP index of grain production in China's main grain producing areas from 2001 to 2017 were compared among the three regions.

## Empirical analysis on the efficiency of grain production

### Time variation characteristics of TFP

Fig 1 shows the TFP index of grain production in China's main grain producing areas from 2001 to 2017. From the overall level, the TFP of grain production in China's main grain producing provinces is relatively low, although there are many positive growth years, the growth rate is very small, and the negative growth years are few, but the decline rate is very large. This is because since 2001, the Chinese government has intensified its financial support for agricultural production by formulating a series of financial policies, improving the policy environment to encourage increased grain production, reducing the agricultural loan interest, as well as increasing the vested interests of peasants and the number of agricultural employees. As a result, farmers' enthusiasm to engage in agricultural production has greatly improved. At the same time, the main grain producing areas have been strictly implementing a farmland protection policy; therefore, so we must put an end to the arbitrary occupation of cultivated land, implement incentive measures, and mobilize the enthusiasm and initiative of rural collective economic organizations and farmers to protect cultivated land. Therefore, the TFP of grain production in China's main grain producing areas has increased over the years. However, due to the asymmetry in agricultural market information and the lack of a perfect market guarantee mechanism, farmers can only plant crops according to the market sales of agricultural products in the previous year, without considering the market demand for the current year, which will seriously affect the future production and consumption of farmers. As a result, the negative growth years of TFP in China's main grain producing areas are relatively few, but the decline is very large. In view of this, the Chinese government should formulate a perfect market security mechanism and increase the visibility of agricultural market information to protect farmers' interests.

From the perspective of time, as shown in Figs 1 and 2, the TFP of grain production in China's main grain producing areas showed an overall fluctuation and gradual decline trend from 2001 to 2017, with an average annual decline of 7.3%. TFP dropped by 1.2% and 8.5% between 2001–2003, this is because before 2003, due to the government's weak financial support, the agricultural production in the main grain-production areas was mostly handled by small-scale family farmers, whose contracted land management area was too small to meet the needs of family production. Thus, traditional small-scale farmers had little enthusiasm for grain production, making it difficult to guarantee grain output. In 2004, it was 4.5% higher than 2003, The reason may be that since 2003, the Chinese government has increased its financial support for agricultural production and formulated corresponding subsidy policies, while main grain

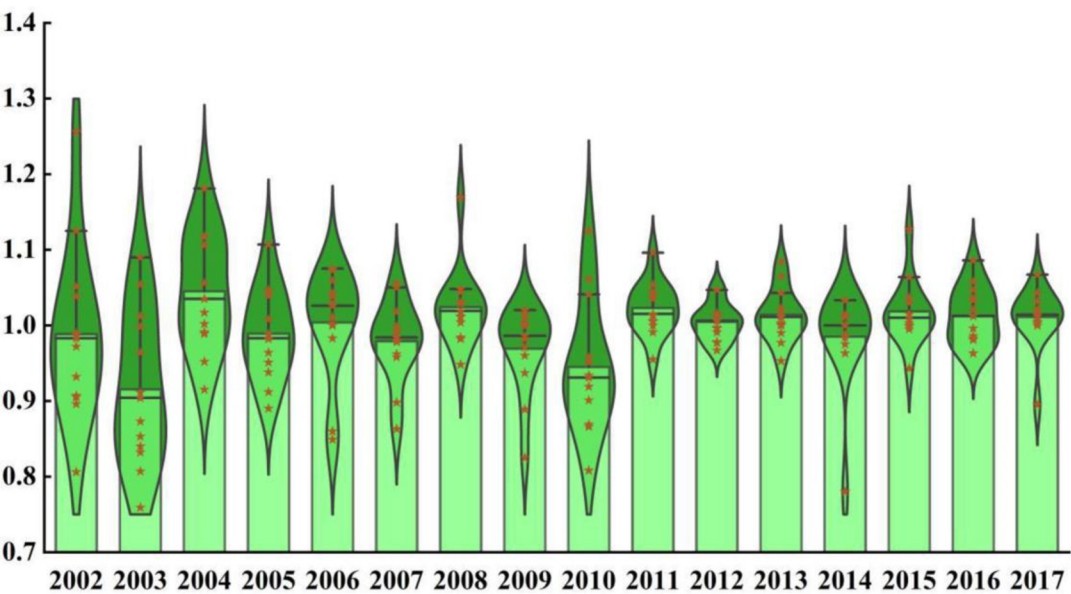

**Fig 1. TFP of grain production from 2001–2017.**

producing provinces have received great support in the form of national subsidies. Farmers in grain producing areas have received many agricultural subsidy funds, and farmers' enthusiasm for agricultural production has greatly increased. The labor force needed for agricultural production has increased substantially, promoting a substantial increase in TFP. However, from 2004 to 2007, except for the small increase in TFP of grain production by 0.4%, it fell by 1.1% and 2.1% in 2005 and 2007, respectively, which is due to the low effective utilization rate of resources—caused by the extensive agricultural management mode formed for a long time— the excessive use of pesticides and chemical fertilizers, and industrial pollution, soil pollution

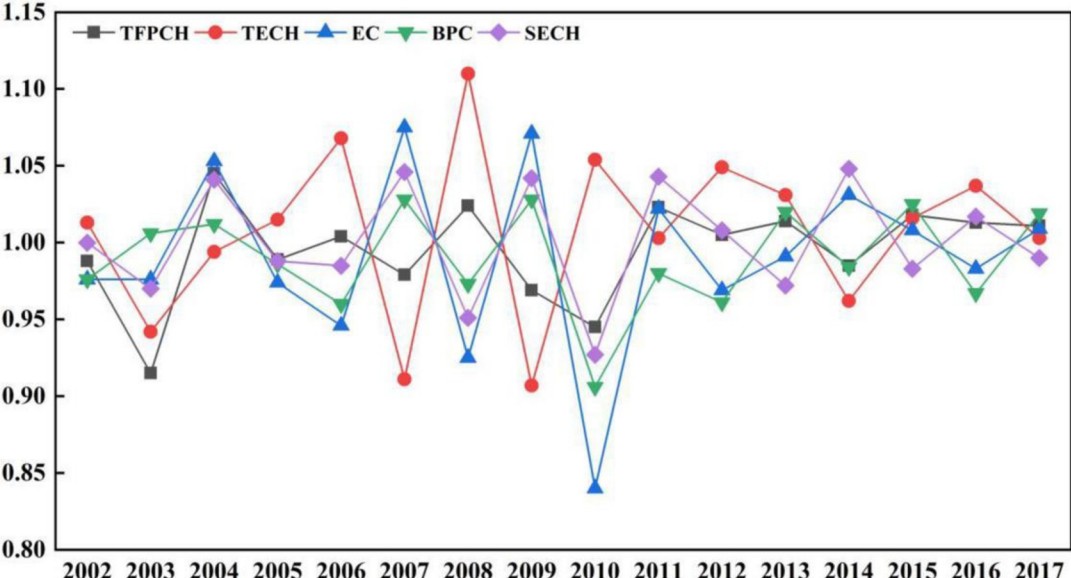

**Fig 2. TFP and decomposition index of grain production from 2001 to 2017.**

and land resource shortages have been increasing. In addition, with the increasing pressure of population expansion, farmers and herdsmen have taken to blindly cultivating star barren hills, at the expense of the ecological environment, to realize higher grain production and income, resulting in a decline in land quality. In 2008, it was 2.4% higher than 2007, The reason may be that the state substantially increased the minimum purchase price of grain in 2008, the vested interests of farmers increased, and the number of people engaged in agricultural production continued to increase. In 2009 and 2010, it dropped by 3.1% and 5.5%, respectively, this is because in 2009 and 2010, there were many natural disasters in China, the drought and flood disasters intensified, and the affected area of cultivated land increased greatly, which led to the reduction of input factors in grain production, thus showing the low performance of low input and low output in grain production. However, from 2010 to 2013, the TFP of grain production in China's main grain producing areas increased by 2.3%, 0.5% and 1.4%, respectively, the reason may be that in 2010, China's main grain producing areas overcame the difficulties in early warning of sudden agricultural disasters, and increased investment in disaster prevention and mitigation infrastructure construction to protect agricultural land from disasters to the greatest extent, so as to ensure the stable supply of grain production. Although TFP declined slightly in 2014, compared with 2013, it dropped by 1.5%, it may be that with the acceleration of urbanization and the continuous improvement in access to agricultural education, farmers now have numerous opportunities to expand; however, many young and middle-aged laborers have migrated from rural to urban areas, resulting in a shortage of rural labor for agricultural production. However, from 2014 to 2017, the TFP of grain production in China's main grain producing areas increased by 1.8%, 1.3% and 1.1%, This is because with the continuous improvement of the level of science and technology, the main grain production areas can increase the training of grain production technology and promote new agricultural production technologies to optimize the level of grain output and efficiency, thereby promoting the level of TFP.

### Decomposition index characteristics of TFP

From the perspective of decomposition index, as shown in Fig 2, the TFP of grain production in China's main grain producing areas showed an overall fluctuation trend from 2001 to 2017, among them, the largest decline was in 2003, which was a decrease of 8.5%, and the largest increase was in 2004, which was an increase of 4.5%. The changing trend of technical efficiency is basically the same as that of TFP. At the same time, technological progress increased by 0.7%, but the growth rate of technical efficiency dropped by 0.9%, indicating that the decline in the TFP index was mainly due to the decline in technical efficiency. In addition, the decline in technical efficiency is mainly due to the decline in pure technical efficiency, with an average annual decline of 1.1% in pure technical efficiency and little change in scale efficiency. The decline in pure technical efficiency shows that there was a disconnection between technology and production in grain production in the main grain production areas from 2001 to 2017, and the phenomenon of different grain production technology channels. Main grain production areas failed to integrate the relevant policies supported by the state's financial resources with the innovation of agricultural production technology, and failed to further promote the construction of irrigation and other water conservancy infrastructure. Meanwhile, the scale efficiency has increased by only 0.1%, which means that the growth dividend of grain production brought about by the scale efficiency is gradually disappearing.

From the decomposition index, as shown in Fig 3, the pure technical change index of the main grain producing areas fluctuated greatly from 2001 to 2017, and the overall level of the sample period was low, with the average value less than or equal to 1. This means that the main

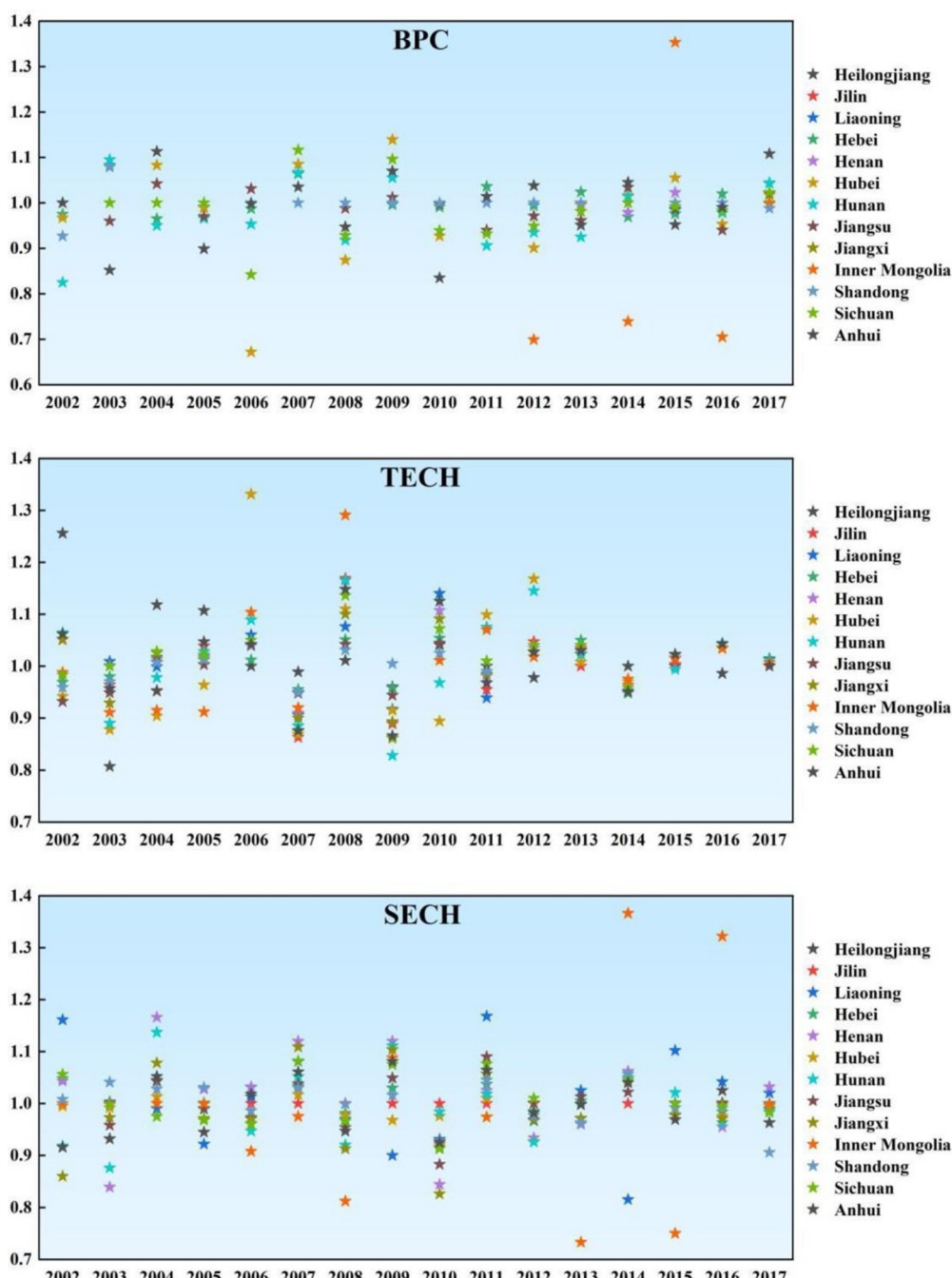

**Fig 3. Decomposition index of TFP of grain production from 2001 to 2017.**

grain producing areas have neither fully integrated the relevant policies with the innovation of agricultural production technology, nor have they promoted the reform of the factor input mechanism. It is, thus, difficult to accelerate the advancement of cutting-edge technology, which makes it difficult to improve. The average value of pure technical efficiency in Hunan and Hubei Province decreased by 8.1% and 2.8%, respectively, which is the biggest decline. This is because in the aspect of financial support, the matching degree of grain production

technology and productivity in Hunan and Hubei Province is not high, grain production is out of touch with science and technology, and there is no reasonable optimization of the proportion of input and output, which leads to a sharp decline in pure technical efficiency. From 2001 to 2017, the scale efficiency index of the main grain producing areas fluctuated slightly, and the overall level was relatively high. Except for Hunan, Hubei, Jiangxi and Anhui provinces, the average scale efficiency of other main grain producing areas is greater than 1. This shows that the main grain producing areas can use land transfers to concentrate abandoned land into the hands of farmers with rich planting experience or higher level of grain mechanization, improve the overall level of grain production mechanization, and realize scale operation. Meanwhile, the grain production efficiency of the main grain-production areas differs depending on their geographical location and economic development. Each main grain production area can give full play to the resource advantages of each region to promote the improvement of scale effect. Jiangxi Province has the largest decline of 1.2%, indicating that Jiangxi Province has not carried out land system reform, has not passed the transfer of management rights and operated on a large scale in grain production. From 2001 to 2017, the overall level of technological progress index of the main grain producing areas is relatively high, and the average value index of the other main grain producing areas is greater than 1, this means that the main grain producing areas are actively promoting scientific research and expanding new space for agricultural development. The main grain producing areas can shift the focus of agricultural production to scientific and technological research, improve the overall quality and comprehensive production capacity by optimizing the structure, innovating varieties, improving unit yield and quality. Meanwhile, the Re-introduction of technology, experiment and demonstration, assimilation and absorption, so that the effects of modern science and technology can be brought into full play, and the contribution rate of scientific and technological progress in food production will be increased, thus promoting the improvement of the level of technological progress in main grain production areas.

## Space distribution of TFP

From the perspective of space distribution, this paper uses ArcGIS to draw the spatial distribution diagram of TFP. Fig 4 shows the TFP index of grain production in 13 main grain producing provinces in China from 2001 to 2017, in which the TFP index of Heilongjiang, Jilin, Liaoning, Hebei, Henan, Shandong is relatively high, with an average value greater than 1. This is because with the support of the government's financial support policy, the six provinces can simplify the distribution of subsidized funds, minimize the risk of interception of subsidized funds, maximize the distribution of subsidized funds to farmers. Meanwhile, it can strengthen the infrastructure construction related to agricultural production, promote the construction of high-standard farmland, consolidate farmland water conservancy facilities, improve he effective irrigation rate and water resources utilization coefficient and actively do a good job in disaster prevention and mitigation. Finally, the TFP index of these six provinces is relatively high. However, the overall level of TFP index in Hubei, Hunan, Jiangsu, Jiangxi, Inner Mongolia Anhui and Sichuan Province is low, with an average value less than 1. The reasons may be that although the six provinces have received strong financial support from the government, the subsidy funds have not been used in the construction of infrastructure, have not improved their own agriculture production methods. At the same time, the government has not strengthened the awareness of the red line of cultivated land, the responsibility of supervision and protection of cultivated land is not clear, and the phenomenon of farmland conversion is becoming more and more serious. As a result, the TFP index of these seven provinces is low.

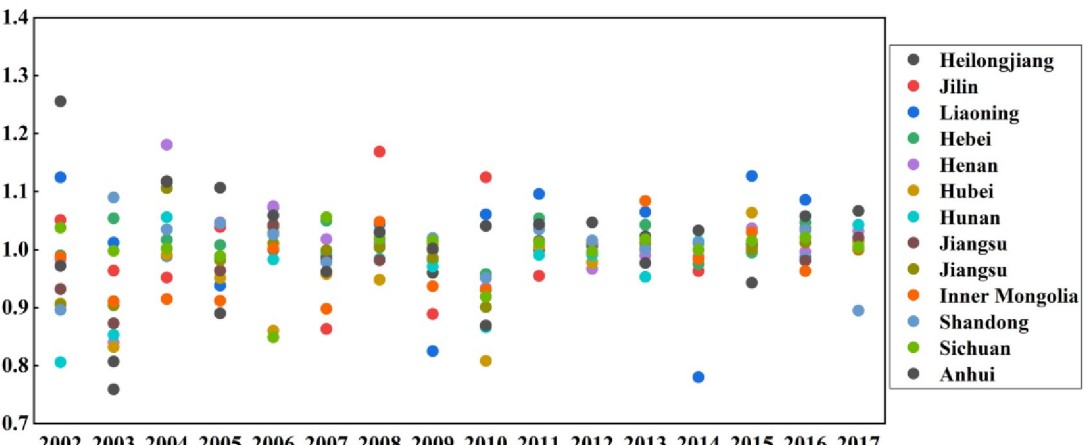

**Fig 4. Provincial distribution of TFP of grain production from 2001–2017.**

Fig 5 intuitively shows the spatial pattern of the TFP of grain production in China's main grain production areas from 2001 to 2017. It can be found that the grain production efficiency in China's main grain producing areas showed the features of uneven spatial development from 2001 to 2017, which generally showed a trend of decline from the central region to the eastern and western regions. From the perspective of regional differences, the central region has the largest difference, followed by the western region and the eastern region has the smallest difference. From 2001 to 2017, there were relatively many provinces with high TFP of grain production, most of which were concentrated in the eastern region, because the eastern regions have not only actively been increasing investment in agricultural scientific research and developing high-yield grain-planting technology but is also ensuring the construction of farmland water reserves and improving the level of agricultural mechanization, thereby ensuring that its grain production is at a high level. From 2001 to 2010, the TFP of grain production in China's main grain producing areas showed a fluctuating downward trend in more provinces. Among them, the TFP of grain production in Hunan Province in the central region decreased to varying degrees in the past nine years. The reason may be that the rainy weather in Hunan Province in the past nine years increased the disaster-stricken area of cultivated land, which directly led to a significant decrease in grain production. From 2011 to 2017, except Shandong Province in the eastern region, the TFP of grain production in other main grain producing areas showed a trend of fluctuation and rise.

In 2002, the TFP of grain production in Heilongjiang Province and Jilin Provinces in the central region and Liaoning Province in the eastern region and Sichuan Province in the western region was greater than 1, the four provincial became a high level of grain production areas. Since these four provinces are rich in arable land resources, the government has actively improved the land protection system, to strengthen the protection of existing arable land. The agricultural land structure should be adjusted, and the area of cultivated land should be reasonably supplemented. In 2007, TFP of grain production in Henan Province in the central region and Hebei Province in the eastern region and Sichuan Province in the western region was greater than 1, the four provincial became a high level of grain production areas. The reason may be that these three provinces dare to innovate, actively develop new land management models, concentrate on planting scale, and improve the moderate scale management system. At the same time, we should strengthen the professionalization of farmers' grain planting, actively organize training courses for grain planting, cultivate farmers' production capacity

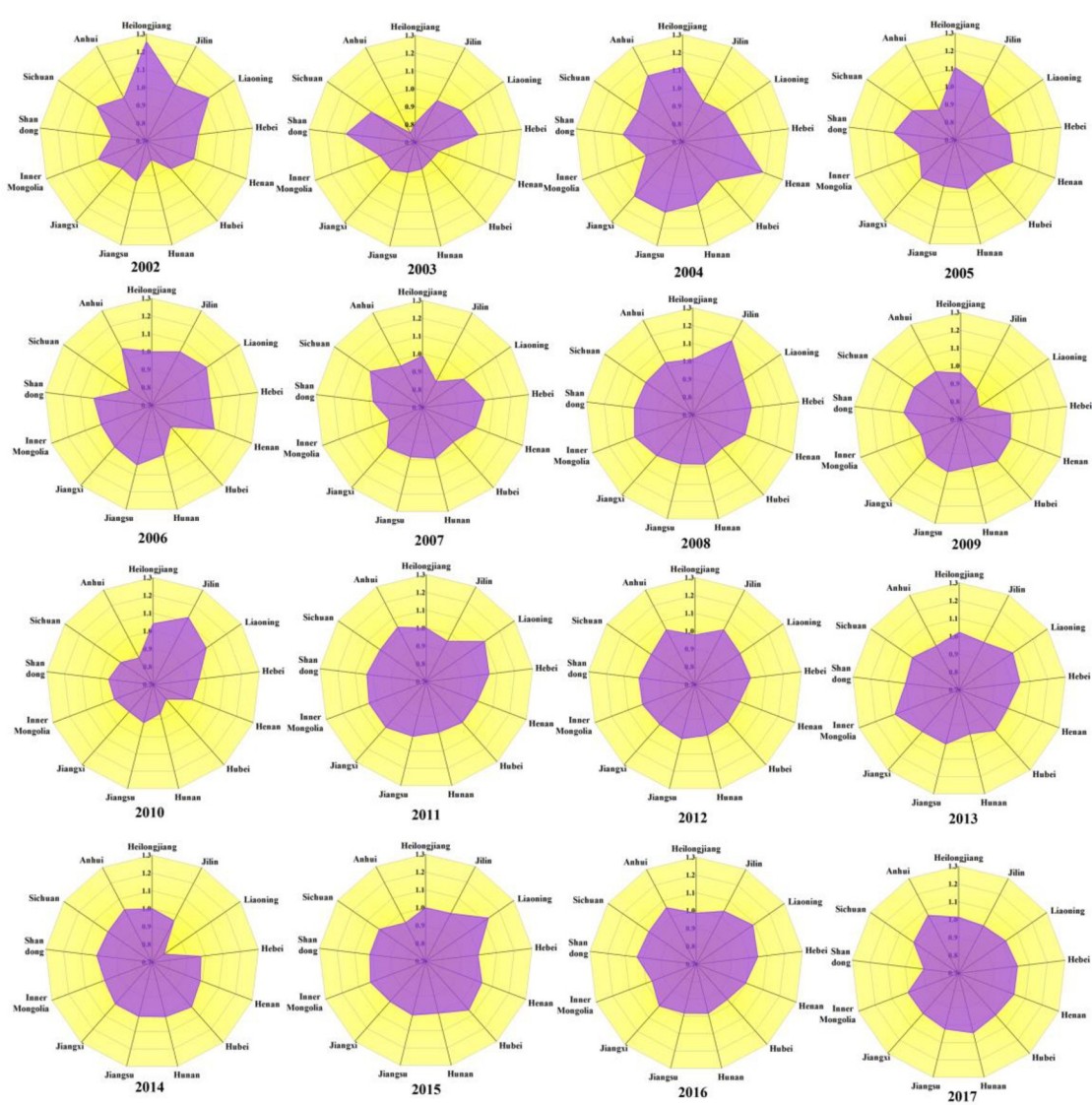

**Fig 5. Spatial distribution of TFP of grain production from 2001 to 2017.**

and management ability in an all-round way, so that farmers can get rid of the traditional planting mode, learn to use scientific and rational methods, improve production efficiency and increase grain production. In 2012, except for Heilongjiang Province, Henan Province, Hubei Province, Hunan Province, Jiangxi Province in the central region and Sichuan Province in the western region, the TFP of grain production in the other main grain producing areas has increased to varying degrees. This means that main grain production areas should increase investment in disaster prevention and mitigation infrastructure construction, continue to carry out cultivated land quality monitoring, increase soil and fertilizer monitoring workstations, real-time monitoring of cultivated land element content, develop practical fertilization technology, reduce blind fertilization, and improve cultivated land quality. In 2017, except for Shandong Province in the eastern region, the TFP of grain production in the other main grain producing areas has increased to varying degrees, The reason may be that with the continuous improvement of the level of science and technology, the main grain producing areas focus on

the cultivation of agricultural science and technology talents, continue to make breakthroughs and innovations in agricultural technology, actively promote the research and development of agricultural science and technology, accelerate the breeding and cultivation of excellent varieties, so that the effects of modern science and technology can be brought into full play, thus promoting the improvement of the TFP level of grain production.

## Conclusions and countermeasures

### Conclusion

The DEA global Malmquist index model was used to calculate the TFP and its decomposition index of grain production in China's main grain producing areas during 2001–2017, and the temporal and spatial differences of grain production efficiency in 13 main grain producing areas in China were analyzed, which provided a solid theoretical basis for boosting the grain production efficiency in China's main grain producing areas. The conclusions are as follows:

(1) From the perspective of time, the TFP of grain production in China's main grain producing areas showed an overall fluctuation and a gradual downward trend from 2001 to 2017, with an average annual decline of 7.3%. Among them, it decreased by 1.2% and 8.5% respectively from 2001 to 2003. Although 2004 was an increase of 4.5% over 2003. However, between 2004 and 2007, except for the small increase of 0.4% in TFP of food production in 2006, it fell by 1.1% and 2.1% in 2005 and 2007 respectively. In 2008, it increased by 2.4% compared to 2007. In 2009 and 2010, it dropped by 3.1% and 5.5% respectively. However, the TFP of grain production in China's main grain producing regions increased by 2.3%, 0.5%, and 1.4% respectively from 2010 to 2013. Although the TFP declined slightly in 2014, down 1.5% from 2013, the TFP of grain production in China's main grain producing areas increased by 1.8%, 1.3%, and 1.1% respectively from 2014 to 2017.

(2) From the perspective of space, the grain production efficiency in China's main grain producing areas from 2001 to 2017 showed an unbalanced spatial development, which was generally manifested as a trend of decline from the central region to the eastern and western regions. From the perspective of regional differences, the central region has the largest difference, followed by the western region and the eastern region has the smallest difference.

(3) From the perspective of decomposition index, the three decomposition indicators of the TFP of grain production in China's main grain production areas are represented by the increase in scale efficiency and technological progress index, the decline in the pure technology change index, and the decline in the TFP index is mainly due to the decline in technical efficiency.

### Countermeasure and suggestion

According to the above empirical results, in order to better improve the efficiency of grain production, we put forward the following four policy recommendations:

(1) Increase financial support for grain production and improve agricultural infrastructure [36]. From a macro point of view, in order to achieve a healthy and stable supply of food production, the government not only needs to invest a large amount of financial funds to support production, but also requires the government to formulate relevant policies to guide it. All main grain producing areas should increase their emphasis on grain

production, actively implement various grain subsidy policies formulated by the state, and abolish various taxes and fees imposed on agricultural production. At the same time, by strengthening the establishment of a monitoring system for natural disasters, paying close attention to observing the situation of diseases, pests and weeds in grain production, and formulating response methods and measures in time to minimize the risk of farmers growing grain and enhance their own comprehensive ability in grain production.

(2) Accelerate the progress of agricultural science and technology and the popularization of agricultural technology, and improve the output efficiency of the food production [37]. All main grain producing areas should vigorously increase investment in scientific research, and earnestly strengthen the promotion of agricultural science and technology. The government should give policy incentives and financial support in terms of special funds for agricultural science and technology promotion and public welfare agricultural technology promotion. At the same time, we should improve the cultivation intensity of new crop varieties, integrate Chinese seed industry resources, rapidly enhance the scientific and technological innovation ability of Chinese seed industry, and increase the coverage rate of improved varieties.

(3) Deepen agricultural reform and achieve scale operations [38]. The main grain producing areas should use the form of land transfer to concentrate the abandoned land into the hands of farmers with rich planting experience or higher level of grain mechanization, improve the overall mechanization level of grain production, and realize scale operation. At the same time, Adopting differentiated policy support measures and modern development strategies [39], it is necessary for the government to adopt targeted and differentiated policy support measures and modern development strategies to promote the improvement of food production efficiency.

## Supporting information

**S1 Table. TFP and decomposition index of grain production from 2001 to 2017.**
(DOCX)

## Acknowledgments

We thank YQ Zhou for useful discussions.

## Author Contributions

**Data curation:** Dehua Zhang.

**Software:** Haiqing Wang.

**Validation:** Sha Lou.

**Writing – original draft:** Shen Zhong.

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
