## [Decision Letter · Decision Letter 0]

18 Jan 2021

PONE-D-20-38013

Research on grain p roduction efficiency  in China ’ s main grain producing areas from the perspective of financial support

PLOS ONE

Dear Dr. Sha Lou

Thank you for submitting your manuscript to PLOS ONE. After careful consideration, we feel that it has merit but does not fully meet PLOS ONE’s publication criteria as it currently stands. Therefore, we invite you to submit a revised version of the manuscript that addresses the points raised during the review process.

We look forward to receiving your revised manuscript.

Kind regards,

Carlos Alberto Zúniga-González, Ph.D

Academic Editor

PLOS ONE

Journal Requirements:

'This work was supported by Heilongjiang Province Philosophy and Social Science Fund Project: Research on the Performance and Influencing Factors of Grain Subsidy Policy in the Agricultural Comprehensive Reform Experimental Area of Liangjiang Plain in Heilongjiang Province (17JYC144).'

'The author(s) received no specific funding for this work.'

4. We note that Figures 1 and 6 in your submission contain map images which may be copyrighted.

a. You may seek permission from the original copyright holder of Figures 1 and 6 to publish the content specifically under the CC BY 4.0 license. 

5. Please ensure that you refer to Figure 1 in your text as, if accepted, production will need this reference to link the reader to the figure.

Additional Editor Comments:

Dear author, I congratulate you on the use of the Malmquist index model with a DEA approach. I consider that the novelty is the variables used to apply the model, so I suggest you take into account the comments of the reviewer 2, and also I suggest one of the last articles (for your references.of the colleague Bravo Ureta and one that is mine:

1) Analysis of the Efficiency of Farming Systems in Latin America and the Caribbean Considering Environmental IssuesWEB OF SCIENCE

Authors: Dios-Palomares, Rafaela; Alcaide, David; Diz, Jose; ... Alberto Zuniga, Carlos; https://www.cabdirect.org/cabdirect/abstract/20153166110
publons.com/p/3106827/

Published: 2015 in Revista Cientifica de la Facultad de Ciencias Veterinarias de la Universidad del Zulia

2) Agricultural productivity growth in Latin America and the Caribbean: an analysis of climatic effects, catch‐up and convergence*

Michée A. Lachaud Boris E. Bravo‐Ureta

First published: 23 December 2020 https://doi.org/10.1111/1467-8489.12408

Reviewers' comments:

Reviewer's Responses to Questions

**Comments to the Author**

1. Is the manuscript technically sound, and do the data support the conclusions?

Reviewer #1: Yes

Reviewer #2: No

2. Has the statistical analysis been performed appropriately and rigorously? 

Reviewer #1: Yes

Reviewer #2: No

3. Have the authors made all data underlying the findings in their manuscript fully available?

Reviewer #1: Yes

Reviewer #2: Yes

4. Is the manuscript presented in an intelligible fashion and written in standard English?

Reviewer #1: No

Reviewer #2: No

5. Review Comments to the Author

Reviewer #1: Review Report

Manuscript ID: PONE-D-20-38013

Manuscript Title: Research on grain production efficiency in China’s main grain producing areas from the perspective of financial support

Using the DEA global Malmquist productivity index model, the paper assesses temporal and spatial differences of grain production efficiency in 13 major grain-producing areas in China. To do so, the paper employs a longitudinal provincial-level dataset with data from 2008-2017. The findings suggest an overall reduction in grain production over the period of 2001 to 2017, with an average annual decline of 0.6% mainly due to technical inefficacies in production. The findings also suggested unbalanced spatial development in grain production across the major grain-producing areas. In light of the empirical results, the study is concluded by providing a number of policy recommendations.

Overall, the manuscript good insights into China’s grain production sector, however there are some issues that need to be addressed.

Comments:

1. Major language issues: The manuscript suffers major language and grammatical problems including improper choice of words, punctations, spelling mistakes, grammatical errors, ambagious and confusing sentences, etc. It does not look like an academic article, as the authors use colloquial language. The manuscript needs to undergo a professional language proof reading stage before publication. Let us give a couple of examples:

(i) Line 44-54: in line 46, the author begins the sentence by “however…” why “however”? The sentence that follows is not a contradiction of the previous sentence. Same in line 49, the sentence starting with “therefore…”, why therefore?. In line 52. What is “mu”, it needs to be spelled out. In line 54, “the overall food supply will be severely tested”, what does this mean?

(ii) Lines 116-138: very poor language. I can’t understand what the authors are trying to communicate.

(iii) Line 139: it reads “in order to make up for the defect of existing research”. Improper language, the existing research is not defect! Line 140, it reads “firstly, on the research angle of view”, I don’t understand what this means.

(iv) Line 140-146: long and confusing sentence with poor and confusing language

(v) Lines 150-151: the article reads “compared with previous research data, the time span is longer and the year is newer”. This does not make any sense. I think the author are trying to say: “in comparison to the previous literature, we use recent data with longer time dimension”.

(vi) Line 159-163. “Kth DMU, f inputs, the input variable is denoted as Xik… g output…….”. This should be kth, f , Xik, g, etc. Proper subscripts and superscripts should be used, equations and parts of the equation should be in italic. Where is the consistency?? What is DMU? Spell out for the first time. Decision-making unit. Same goes for lines 191-193.

(vii) Line 185: “that’s going to be…” should be “that is…”

(viii) Line 258-275: Bad English. In Line 259 “…arising from peasant household agriculture production”. I don’t understand this. Line 262-267, does not relate there, “… we should put an end to the phenomenon of arbitrary occupation…”, the authors give recommendations here under the “empirical analysis section”

(ix) Line 270: what is armers?

(x) Line 295: “… have not improved their own agriculture production methods”. Firstly, bad English. Secondly, based on what evidence the authors claim this? Provide a citation and explain why.

(xi) Line 300: “…productivity index of these six provinces is not high.”, what does “not high” imply? what is the limit of high? Bad English. Same mistake in line 254, what is “relatively not high”?

(xii) Line 318: what is ain?

(xiii) Line 323, and 340: Punctations issues: put comma before the word “respectively”

(xiv) Line 338: “thus showing the low performance of low input and low output in grain production”. Ambiguous and confusing. I can’t understand this.

(xv) Line 368, “disconnect” should be “disconnection”

These are only a few examples that I wanted to point out to. The entire manuscript has English language problems.

2. The manuscript is poorly structured. Long and confusing sentences, paragraphs , and sections, lack of sub-headings etc.

(i) Line 80-84: While, the authors discusses the presentation of the paper: “the second part introduces….. the third part introduces…”, the “parts” are unowned and unnumbered. As a reader I do not understand which is part 4 and which is part 2. Also they are not parts, they are sections.

(ii) Under the sub-heading in line 294 “time variation characteristics of TPF and its deco composition index of grain production in main grain producing areas”, the section is nearly 9 pages long! This too long and confusing for the readers, as one could easily lose focus. Also, might I suggest using shorter sub-headings. The sub-heading in line 245 is extremely long, for instance you don’t need to write “Main grain producing areas”, the article makes that clear in the previous sections.

(iii) The sub-section in 423 is redundant and repetitive. This section repeats the discussion presented in lines 279 – 300”. The two sections disusing geographical/spatial dimensions of grain productivity should be combined. This makes the manuscript unnecessarily too long and repetitive.

(iv) Another example of poor structuring: line 263, the authors provide recommendations under the “empirical strategies section”

(v) Stay consistent in using abbreviations throughout the article. In line 249 the abbreviation of TFP is used, it is never spelled out in earlier pages. Once TFP is used, continue using TPF in the following pages, instead of “Total Factor Productivity” for maintaining consistency.

(vi) No page numbers.

3. The introduction section is not clear and does not outline the problem, scope of the problem, research objective and questions. It also lacks coherency. The authors provide present in different scales, making it hard to compare. For instance, the aggregate grain output 617.91 tons, but consumption is 479 kg per person. It would be better to calculate production per capita, so one could compare it with per capita consumption.

4. The literature review discusses different estimation approaches but does not provide justification for the preferences of the DEA method. Why is DEA the best method? For instance, in line 164, the approach assumes CRS. In reality, the CRS property of the production function may not hold. Moreover, the DEA does not allow to separate random error from inefficacy, etc. see for example

Wadud, A., White, B., 2000. Farm household efficiency in Bangladesh: a comparison of stochastic frontier and DEA methods. Applied Economics. 32, 1665–1673

5. The other limitation comes from data. Using provincial level data does not capture micro- or farm-level variations. Total population of a province is used as labour input in production. The findings are indicative at best. Hence the authors need to clearly write limitations of the study, especially limitations attributable to data. This is the major empirical short-coming in the paper, so it needs to stand out.

6. Line 282 and 293. The total factor productivity index greater and smaller than 1. The author needs to clearly define and interpret the index, discuss the min and max range of the index and what they signify. Does higher index signify more productivity? Why the value 1 is regarded as a benchmark? Discuss and clarify to help the reader better understand the estimated ranges of the index.

Reviewer #2: Reviewer Report on “Grain Production Efficiency in China’s main grain producing areas from the perspective of financial support” submitted to PLOS ONE

The paper under review analyzes an important but not very novel issue, estimating grain production efficiency in China with financial support as an input. While I think the issue is worth examining, the data and method employed are both inadequate to provide convincing answers to the research questions posed, which seriously undermines the scientific value of this paper. My specific comments are outlined below in greater detail.

1. My biggest concern is the lack of innovation in this paper. The mere “contribution” of this paper is the addition of a financial support indicator in the efficiency analysis. The authors argued (in lines 141-143) that “… most of the literature [is] based on the three aspects of select[ed] indicators [l]abour, land and capital…”. (Despite the many grammatical errors in this sentence) this argument is definitely not true. Previous studies have examined factors such as land fragmentation (Nguyen et al., 1996; Feng, 2008), political violence (Gonzalez and Lopez, 2007), labor migration (Mochebelele and Winter-Nelson, 2000; Wouterse, 2010), and household income sources (Zhong et al., 2019), to name a few. In fact, the inclusion of financial factors is also not new—see, e.g., Bojnec and Latruffe (2011) for an early study. The application of a DEA-based method with a handful of input variables is also far from innovative (more on this point below).

References:

Bojnec, Š., Latruffe, L. (2011). Financing availability and investment decisions of Slovenian farms during the transition to a market economy. Journal of Applied Economics, 14(2), 297-317.

Feng, S. (2008). Land rental, off-farm employment and technical efficiency of farm households in Jiangxi Province, China. NJAS – Wageningen Journal of Life Sciences, 55(4), 363-378.

Gonzâlez, M., Lopez, R. (2007). Political Violence and Farm Household Efficiency in Colombia. Economic Development and Cultural Change 55(2), 367-392.

Mochebelele, M.T. and Winter-Nelson, A. (2000), “Migrant labor and farm technical efficiency in Lesotho”, World Development, 28(1), 143-153.

Wouterse, F. (2010), “Migration and technical efficiency in cereal production: evidence from Burkina Faso”, Agricultural Economics, 41(5), 385-395.

Zhong, M., Zhu, Y., Chen, Q., Liu, T., Cai, Q. (2019). Does household engagement in concurrent business affect the farm size–technical efficiency in grain production? Evidence from northern China. China Agricultural Economic Review 11(1), 125-142.

2. The authors seem to mix up the concept of “total factor productivity” with that of “production efficiency”. The former refers to the “overall” contribution of technological progress in the process of economic growth, while the latter refers to the distance of each production unit to the “given” production frontier – they measure different things at different scales. Put differently, even if the production frontier is fixed, you can still estimate production efficiency (for each of the production units), but you cannot estimate total factor productivity in that csae.

3. The data are not well-suited for the intended analysis. The analysis was based on provincial data. The big question is: Can a province serve as a DMU (decision-making unit) in a production problem? Who is making production-related decisions (involving input choices, management, and even land transfers)? The farmers! Not the provinces. I would say in the 1990s, when farm household data were not readily available, the use of provincial data may be OK; at least, they may reveal some general patterns. However, with tons of farm-level data available in China today, the analysis conducted at the provincial level is largely uninformative, and potentially misleading even. I suspect that the aggregation of input and output data from the farm level to the provincial “kills” a lot of efficiency loss because the averaging exercise tends to drug down the production frontier within a given province.

4. The methodology is not sound. There are two major problems. First, the input variables are limited. How about irrigation conditions? How about natural disasters? They all matter for grain production. It’s well-known that the lack of input data will lead to misleading efficiency estimates. This problem is likely to occur in the paper under review.

Second, it’s unclear what role financial support plays in the story. The author treats it as an input in the DEA analysis, but how would loans directly affect agricultural outputs (—I don’t think throwing money in the field would make crops grow)? In theory, it plays the role of releasing the budget constraints of the farmers. Thus, it affects production efficiency mainly through improving “allocative efficiency”, which, however, is not examined in the paper. See, e.g., Smith et al. (2011) for a study examining allocative efficiency.

Reference:

Smith, R. B. W., Gemma, M., & Palinisami, K. (2011). Profit based efficiency measures, With an application to rice production in Southern India. Journal of Agricultural Economics, 62(2), 340-356. https://doi.org/10.1111/j.1477-9552.2010.00288.x

5. There are many language problems in the paper, making the paper rather frustrating to read. The authors should check the write-up carefully. Some obvious examples include:

1) “…the previous literature is not only the time span is short” Which is the subject of the sentence, “the literature” or “the time span”?

2) Lines 139-140: “…the innovation points of this paper is mainly in the following three aspects” (problem: Disagreement between the subject and the verb) should be “…the innovation of this paper is mainly in the following three aspects” or “the innovation points of this paper are mainly in the following three aspects”.

6. PLOS authors have the option to publish the peer review history of their article (what does this mean?). If published, this will include your full peer review and any attached files.

Reviewer #1: **Yes: **Hayatullah Ahmadzai

Reviewer #2: No

---

## [Author Response · Author response to Decision Letter 0]

8 Feb 2021

Dear Editors and Reviewers,

Thank you for your letter and the reviewer’s comments concerning our manuscript entitled “Research on grain production efficiency in China’s main grain-producing areas from the perspective of financial support” (PONE-D-20-38013). Those comments are all valuable and very helpful for revising and improving our paper, as well as the important guiding significance to our researches. We have studied comments carefully and have made correction which we hope meet with approval. Revised portion are marked in red in the paper. The main corrections in the paper and the responds to the reviewer’s comments are as flowing. 

Responds to the Journal’s comments:

1 Please ensure that your manuscript meets PLOS ONE’s style requirements, including those for file naming. 

The author’s answer: 

We fully respect the requirements of the journal and we have modified the format of the article according to the requirements of the journal, hoping to meet the PLoS One’s style requirements.

2 We suggest you thoroughly copyedit your manuscript for language usage, spelling, and grammar. If you do not know anyone who can help you do this, you may wish to consider employing a professional scientific editing service

The author’s answer: 

We fully respect the requirements of the journal and we have polished part of the the manuscript in the editage organization cooperating with PLoS One, and revised the language and grammar problems in the manuscript. The order screenshot is as follows:

3 The question in the acknowledgments section.

The author’s answer: 

We fully respect the requirements of the journal and the funding information should not appear in the acknowledgement section or other areas of the article. Therefore, we will apply for the journal to remove any funding related words from the manuscript on our behalf in the cover letter by deeply greatful. We have removed any funding related words in the revised manuscript. Meanwhile, because of the author’s negligence, we didn’t fill in the fund project in the process of submission. We will reinsert the funding information in the process of submitting the revised manuscript. The authors received specific funding for this work, funding information are as follows:

This work was supported by Heilongjiang Province Philosophy and Social Science Fund Project (18JYE669); Youth Innovative Talents Training Program in General Undergraduate Colleges and Universities in Heilongjiang Province (UNPYSCT2018125); Special support project for postdoctoral in Heilongjiang Province (LBHZ19073)；Special plan for top young talents of Harbin University of Commerce (2020CX42); Graduate Innovation Project of Harbin University of Commerce (YJSCX2020-643HSD).

4 We note that Figures 1 and 6 in your submissions contain map images which may be copyrughted. 

The author’s answer: 

We fully respect the requirements of the journal and as for the copyright protection of the maps in the manuscript, we have completely deleted the map in Figure 1 in the manuscript, and changed the map shown in Figure 6 in the empirical analysis to radar map, which may highlight the differences between regions

5 Please ensure that you refer to Figure 1 in your text as, if accepted, production will need this reference to link the reader to the figure

The author’s answer: 

We fully respect the requirements of the journal and we have completely removed the map in Figure 1 from the manuscript

Responds to the additional editor’s comments:

1 I suggest you take into account the comments of the reviewer 2, and also I suggest one of the last articles for your references.of the colleague Bravo Ureta and one that is mine:

The author’s answer: 

We humbly accepted the comments of the additional editor, fully considered the comments of reviewer 2 and a new input variable has added to the paper. Meanwhile, we fully referred to the two articles of the additional editor and his colleague Bravo ureta, and quoted the two articles of the additional editor and his colleague Bravo ureta into this manuscript. The two articles are as follows:

(1)Rafaela Dios-Palomares, Dacid Alcaide, Jose Diz, Manuel Jurado, Angel Prieto, Martina Morantes, Carlos Alberto Zuniga. Analysis of the Efficiency of Farming Systems in Latin America and the Caribbean Considering Environmental Issues. Revista Cientifica. 2015: 43-50.

(2)Michée A. Lachaud, Boris E. Bravo‐Ureta, Carlos E. Ludena. Agricultural productivity growth in Latin America and the Caribbean: an analysis of climatic effects, catch‐up and convergence*. International Conference of Agricultural Economists. 2020: 1-59.

Responds to the reviewer’s comments:

Reviewer 1:

1 Major language issues: The manuscript suffers major language and grammatical problems including improper choice of words, punctations, spelling mistakes, grammatical errors, ambagious and confusing sentences, etc. 

The author’s answer: 

We fully respect the requirements of the reviewer 1. On the whole, we have polished part of the manuscript in the Editage organization cooperating with PLoS One, and revised the language and grammar problems existing in the manuscript. Next, we will answer the language and grammar questions raised by reviewer 1 one by one

(1)Line 44-54: in line 46, the author begins the sentence by “however…” why “however”? The sentence that follows is not a contradiction of the previous sentence. Same in line 49, the sentence starting with “therefore…”, why therefore? In line 52. What is “mu”, it needs to be spelled out. In line 54, “the overall food supply will be severely tested”, what does this mean?

The author’s answer: 

The sentence after "however" does not contradict the sentence before, so we change "however" to "meanwhile".

There is no connection between the following sentence and the previous one. We have deleted the conjunction therefore from the manuscript.

"Mu" refers to the unit of land. After carefully looking up the data, it is found that other scholars' English translations are also mu, and there is no need to spell it out.

The sentence "the overall food supply will be severely tested" indicates: as mentioned above, China's cultivated land supply has been insufficient and will show a decreasing trend in the future, which will directly lead to the shortage of grain supply, so the overall grain supply will be severely tested.

(2)Lines 116-138: very poor language. I can’t understand what the authors are trying to communicate.

The author’s answer: 

This section mainly discusses the second aspect of the research on grain production efficiency in the literature review part of this manuscript, which is the index and data selection for measuring grain production efficiency. Combined with previous studies, this part successfully discusses the index and data selection for measuring grain production efficiency in four authors' articles. We have polished part of the manuscript in Editage organization cooperating with PLoS One. I believe this paragraph will be clearer after polishing.

(3)Line 139: it reads “in order to make up for the defect of existing research”. Improper language, the existing research is not defect! Line 140, it reads “firstly, on the research angle of view”, I don’t understand what this means.

The author’s answer: 

We have changed "in order to make up for the defect of existing research" to "compared to previous studies"

"First, on the research angle of view" the reviewer doesn't understand it because we don't express it clearly. We mainly want to express "first, in terms of data selection", so we changed "First, on the research angle of view" to "first, in terms of data selection"

(4)Line 140-146: long and confusing sentence with poor and confusing language.

The author’s answer: 

These six lines mainly introduce the two innovations of this paper. The first is to introduce the index of agricultural loans into the evaluation system of grain production efficiency, highlighting the importance of financial support for grain production in China's major grain producing areas. Secondly, based on the classic DEA model, the global Malmquist index model is constructed to evaluate the grain production efficiency. We have polished part of the manuscript in the editage organization cooperating with PLoS One. We believe that the sentences after polishing will not be lengthy and messy, and the language will not be confused. 

(5)Lines 150-151: the article reads “compared with previous research data, the time span is longer and the year is newer”. This does not make any sense. I think the author are trying to say: “in comparison to the previous literature, we use recent data with longer time dimension

The author’s answer: 

According to the reviewers, we have changed "compared with previous research data, the time span is longer and the year is new" to "in comparison to the previous literature, we use recent data with longer time dimension"

(6)Line 159-163. “Kth DMU, f inputs, the input variable is denoted as Xik… g output…….”. This should be kth, f, Xik, g, etc. Proper subscripts and superscripts should be used, equations and parts of the equation should be in italic. Where is the consistency?? What is DMU? Spell out for the first time. Decision-making unit. Same goes for lines 191-193

The author’s answer: 

According to the reviewers, we have changed “Kth DMU, f inputs, the input variable is denoted as Xik… g output…….” to “kth, f, Xik, g, etc”. 

Meanwhile，equations and parts of the equation has been made in italic. 

DMU (decision making unit) has spelled out the whole process when it first appeared.

Lines 191-193 has also been modified, the formula is changed to italics, so that all the formulas are consistent.

(7)Line 185: “that’s going to be…” should be “that is…”

The author’s answer: 

According to the reviewer's opinion, we have changed "that's going to be..." to "that is..."

(8)Line 258-275: Bad English. In Line 259 “…arising from peasant household agriculture production”. I don’t understand this. Line 262-267, does not relate there, “… we should put an end to the phenomenon of arbitrary occupation…”, the authors give recommendations here under the “empirical analysis section”

The author’s answer: 

“…arising from peasant household agriculture production”: we want to express that Chinese government has intensified its financial support for agricultural production by formulating a series of financial policies, improving the policy environment to encourage increased grain production, reducing the agricultural loan interest, as well as increasing the vested interests of peasants and the number of agricultural employees This may be due to the confusion of language that misled the reviewers. This situation will change after the article language is polished in Editage organization.

“… we should put an end to the phenomenon of arbitrary occupation…”: this mainly refers to the main grain producing areas to strictly implement incentive measures, and mobilize the enthusiasm and initiative of rural collective economic organizations and farmers to protect cultivated land. It is mentioned in the original that the reviewers may not read the manuscript clearly due to the language confusion. I believe that this situation will change after part of the manuscript is polished in Editage organization cooperating with PLoS One

We have put the “In view of this, the Chinese government should formulate a perfect market security mechanism, and increase the publicity of agricultural market information to protect the interests of farmers” into the section”4.1 Time variation characteristics of TFP” The "empirical analysis section" mentioned by the author has been listed above.

(9)Line 270: what is armers?

The author’s answer: 

This is our spelling mistake. What we want to represent here in the manuscript is farmers.

(10)Line 295: “… have not improved their own agriculture production methods”. Firstly, bad English. Secondly, based on what evidence the authors claim this? Provide a citation and explain why.

The author’s answer: 

This is a problem of poor English language. We have polished part of the manuscript in Editage organization cooperating with PLoS One. I believe the language after polishing will not be confused.

As shown in the following, relevant scholars believe that changing the mode of agricultural production can improve the efficiency of grain production at the micro level. Therefore, we believe that the mode of agricultural production is an important factor affecting the efficiency of grain production. After careful review of the data, it is found that other scholars also believe that there are problems in the agricultural production mode in the main grain producing areas. Therefore, we believe that the total factor productivity index of Hubei, Hunan, Jiangsu, Jiangxi, Inner Mongolia, Anhui and Sichuan provinces is relatively low, the reason may be that “… have not improved their own agriculture production methods”.

(1)Erin M, Tegtmeier and Michael D and Duffy. External Costs of Agricultural Production in the United Atates. International Journal of Agricultural. 2014.2(1):1-20.

(2)Monchi Lio, Meng-Chun Liu. Governance and agriculture productivity: A cross national analysis. Food Policy 33 (2008) 504-512.

(3)Mingyang Sun, Jieming Chou, Yuan Xu, Fan Yan, Jiangnan Li. Study on the thresholds of grain production risk from climate change in China’s main grain producing areas. Physics and Chemistry of the Earth 116(2020) 102837.

(4)Wang peng, Deng xiangzheng, Jiang sijian. Global warming, grain production and its efficiency: Case study of major grain production region. Ecological Indicators 105 (2019) 563-570. 

(11)Line 300: “…productivity index of these six provinces is not high.”, what does “not high” imply? what is the limit of high? Bad English. Same mistake in line 254, what is “relatively not high”?

The author’s answer: 

“… The value of "not high" in productivity index of these six productions is not high is less than 1. We also changed "not high" to "low".

In the calculation of total factor productivity, as well as the previous literature research, the total factor productivity value is compared with 1. If it is smaller than 1, it means that the total factor productivity level is low; if it is larger than 1, it means that the total factor productivity level is high.

what is “relatively not high”? It refers to the low level of total factor productivity in China's major grain producing areas. We also changed the wrong language "relatively not high" to "low".

(12)Line 318: what is ain?

The author’s answer: 

Ain is our misspelling, redundant word, which we have removed from the manuscript.

(13)Line 323, and 340: Punctations issues: put comma before the word “respectively”.

The author’s answer: 

According to the reviewer's opinion, we have added comma before each word "perspective" from line 323 to 340.

(14)Line 338: “thus showing the low performance of low input and low output in grain production”. Ambiguous and confusing. I can’t understand this.

The author’s answer: 

As mentioned above, there were many natural disasters in China in 2009 and 2010, the aggravation of drought and flood disasters, and the large increase of farmland affected area, which led to the reduction of input factors in food production. According to the input-output theory, the reduction of input factors would inevitably lead to the reduction of output, so "thus showing the low performance of low input and low output in grain production". At the same time, we have polished part of the manuscript in the editage organization cooperating with PLoS One. I believe that the sentences after polishing will make the readers understand clearly.

(15)Line 368, “disconnect” should be “disconnection”

The author’s answer: 

As requested by the reviewers, we have changed "disconnect" to "disconnect".

2 The manuscript is poorly structured. Long and confusing sentences, paragraphs, and sections, lack of sub-headings etc.

We fully respect the requirements of the reviewer. First of all, on the whole, we have embellished part of the manuscript in the editorial organization cooperating with PLoS One, and revised the language problems existing in the manuscript. Next, we will reply to reviewer 1's questions about the structure of the article one by one.

(1)Line 80-84: While, the authors discusses the presentation of the paper: “the second part introduces….. the third part introduces…”, the “parts” are unowned and unnumbered. As a reader I do not understand which is part 4 and which is part 2. Also they are not parts, they are sections.

The author’s answer: 

"Parts" is ownerless and numbered, so we change "parts" into "chapter", which becomes "the second chapter introduces .. the third chapter introduces…” In this way, the readers will understand the order of the following text.

(2)Under the sub-heading in line 294 “time variation characteristics of TPF and its deco composition index of grain production in main grain producing areas”, the section is nearly 9 pages long! This too long and confusing for the readers, as one could easily lose focus. Also, might I suggest using shorter sub-headings. The sub-heading in line 245 is extremely long, for instance you don’t need to write “Main grain producing areas”, the article makes that clear in the previous sections.

The author’s answer: 

After carefully looking up the data and studying the articles of other scholars, we have divided the title "time variation characteristics of TPF and its Decomposition index of grain production in main grain producing" into "from the perspective of time" and "from the perspective of decomposition index" for analysis, and added as a subtitle, so that readers will not be confused and will not lose focus. According to the reviewer's suggestion, we delete the words "main grain producing areas" in the subtitle of the empirical part.

(3)The sub-section in 423 is redundant and repetitive. This section repeats the discussion presented in lines 279-300”. The two sections discussing geographical spatial dimensions of grain productivity should be combined. This makes the manuscript unnecessarily too long and repetitive.

The author’s answer: 

We have added the empirical analysis of lines 279-300 to" 4.3 space distrubition of TFP ". At the same time, we modified and embellished the manuscript to make it unnecessary to be too long and repetitive.

(4) Another example of poor structuring: line 263, the authors provide recommendations under the “empirical strategies section”

The author’s answer: 

The reviewer's question is consistent with the eighth small question in the first big question language question. We have answered the eighth small question in the first big question language question of the reviewer. The answers are as follows: We have put the “In view of this, the Chinese government should formulate a perfect market security mechanism, and increase the publicity of agricultural market information to protect the interests of farmers” into the section ‘4.1 Time variation characteristics of TFP”. The "empirical analysis section" mentioned by the author has been listed above.

(5) Stay consistent in using abbreviations throughout the article. In line 249 the abbreviation of TFP is used, it is never spelled out in earlier pages. Once TFP is used, continue using TPF in the following pages, instead of Total Factor Productivity” for maintaining consistency.

The author’s answer: 

In addition to the "total factor productivity" in the introduction, we have changed all the "total factor productivity" in the manuscript to "TFP", which can make it consistent in the article.

(6) No page numbers.

The author’s answer: 

We have added page numbers to the full text.

3 The introduction section is not clear and does not outline the problem, scope of the problem, research objective and questions. It also lacks coherency. The authors provide present in different scales, making it hard to compare. For instance, the aggregate grain output 617.91 tons, but consumption is 479 kg per person. It would be better to calculate production per capita, so one could compare it with per capita consumption.

The author’s answer: 

First, the introduction of the manuscript outlines the problems of China's food security, the scope of which is the low rate of food self-sufficiency and the insufficient supply of arable land. 

Second, at the same time, the purpose of the research is added in the introduction. That is “In this context, it is particularly important to improve the grain production efficiency of China's major grain producing areas3”

According to the reviewer's opinion, in order to maintain consistency, the per capita food production and per capita food consumption were compared. We have put “The per capita grain production is 472kg”into section “Introduction”. That is “According to the 2018 China Grain Development Report released by the China Food and Strategic Reserves Administration, China’s total grain output in 2017 was 617.91 million tons, with the per capita grain production being 472 kilograms. Meanwhile, China’s per capita grain consumption is 479 kilograms, making the self-sufficiency rate of grain only 82.3%, which is lower than the world security standard of 90%”. .

4 The literature review discusses different estimation approaches but does not provide justification for the preferences of the DEA method. Why is DEA the best method? For instance, in line 164, the approach assumes CRS. In reality, the CRS property of the production function may not hold. Moreover, the DEA does not allow to separate random error from inefficacy, etc. see for example

The author’s answer: 

In view of the suggestions put forward by the reviewers and the classic references and the classical literature we consulted, we have listed the answers to "why is DEA the best method?"

Firstly, DEA method is used to evaluate the production (operation) performance of multi input and multi output decision-making unit. DEA method does not need to specify the form of input-output production function, and can evaluate the efficiency of decision-making unit (DMU) with complex production relations.

Secondly, it has the characteristic of unit invariance, that is, the results of DMU measured by data envelopment analysis (DEA) are not affected by the unit selected by input and output data. As long as the unit of input and output data is unified, any change of input and output data unit will not affect the efficiency result. It can process both proportional data and non proportional data at the same time, that is, it can use both proportional data and non proportional data in input and output data, as long as the data is the main indicator reflecting the input or output of decision-making units.

Thirdly, the weight of the model in DEA is generated by mathematical programming based on the data, which does not need to set the weight of input and output in advance and is not affected by human subjective factors. Expert evaluation and other pre-set weight methods are easily affected by human subjective factors.

Fourthly, DEA is used to compare the target value with the actual value, sensitivity analysis and efficiency analysis. It can further understand the resource utilization of decision-making units and provide reference for management decision-making.

We have made a brief modification and added it to the last paragraph of chapter "literature review", that is “The strengths of DEA is that it is nonparameteric and there is no need to make these assumptions. It also allows for multiple outputs and inputs and the DEA does not allow to separate random error from inefficacy”. Meanwhile, the classic literature provided by reviewers is also cited in the article.

(1)Wadud, A., White, B., 2000. Farm household efficiency in Bangladesh: a comparison of stochastic frontier and DEA methods. Applied Economics. 32, 1665–1673.

(2)Cooper W W, Seiford L M, Tone K. Data envelopment analysis: a comprehensive text with models, applications, references and DEA-Solver software[M]. 2nd ed. New york: Springer Science & Business Media, 2007.

(3)A.Boussofiane, R.G. Dyson and E. Thanassoulis. Applied data envelopment analysis. European Journal of Operational Research 52 (1991) 1-15.

(4)Kaoru Tone. A slacks-based measure of efficiency in data envelopment analysis. European Journal of Operational Research 130 (2001) 498-509.

5 The other limitation comes from data. Using provincial level data does not capture micro- or farm-level variations. Total population of a province is used as labour input in production. The findings are indicative at best. Hence the authors need to clearly write limitations of the study, especially limitations attributable to data. This is the major empirical short-coming in the paper, so it needs to stand out.

The author’s answer: 

We agree with the reviewers that there are indeed restrictions on data selection, using provincial level data does not capture micro- or farm-level variations. However, the samples studied in this paper are 13 major grain producing areas in China, which are divided by provincial units. 

At the same time, the index data published by the websites and statistical yearbooks of China Bureau of statistics and regional statistical bureaus are not unified, only the provincial data are unified, and there is no unified branch for farm level data. Therefore, in order to ensure the standardization and accuracy of the data, we choose the provincial data to study. We have added it to section "3.2 index selection, sample selection and data sources"

6 Line 282 and 293. The total factor productivity index greater and smaller than 1. The author needs to clearly define and interpret the index, discuss the min and max range of the index and what they signify. Does higher index signify more productivity? Why the value 1 is regarded as a benchmark? Discuss and clarify to help the reader better understand the estimated ranges of the index.

The author’s answer: 

These problems are clearly explained in Chapter 8 of Cheng Gang's book.

《Data Envelopment Analysis: Methods anandMaxDEA Software》

The definition of total factor productivity: The comprehensive productivity of production unit as each factor in the system and the portion of output not explained by the amount of inputs used in production.

R Färe, Grosskopf, Lindgren, and Roos (1992) first used DEA method to calculate Malmquist index. In the output results of maxdea software, whether it is input-oriented or output oriented, the meaning of Malmquist index is that greater than 1 means higher productivity and less than 1 means lower productivity.

We also modified it in the paper to explain and explain it in more detail, and added it to section "3.1 DEA Global Malmquist model".

Reviewer 2:

1.My biggest concern is the lack of innovation in this paper. The mere “contribution” of this paper is the addition of a financial support indicator in the efficiency analysis. The authors argued (in lines 141-143) that “… most of the literature [is] based on the three aspects of select[ed] indicators [l]abour, land and capital…”. (Despite the many grammatical errors in this sentence) this argument is definitely not true. Previous studies have examined factors such as land fragmentation (Nguyen et al., 1996; Feng, 2008), political violence (Gonzalez and Lopez, 2007), labor migration (Mochebelele and Winter-Nelson, 2000; Wouterse, 2010), and household income sources (Zhong et al., 2019), to name a few. In fact, the inclusion of financial factors is also not new—see, e.g., Bojnec and Latruffe (2011) for an early study. The application of a DEA-based method with a handful of input variables is also far from innovative (more on this point below).

The author’s answer: 

First of all, for the author's "the authors assigned (in lines 141-143) that" most of the literature [is] based on the three aspects of select[ed] indicators [l]abour, land and capital…” (desire the many graphical errors in this sentence) "we have polished part of the manuscript in Editage organization cooperating with PLoS One. We believe that there will be no serious language problems in the polishing.

Secondly, the research of DEA has strict restrictions on the selection of input and output indicators, because DEA, as a non parametric frontier analysis method, has relatively less requirements on the number of DMUs. If the number of DMUs is less than the number of input-output indicators, it is easy to get the result that most DMUs are effective. Generally speaking, the number of DMUs should not be less than the product of the number of input and output indicators. However, in practical application, the number of DMUs is often fixed, which can only be improved by less input or output indicators. Therefore, we must find the core variable as the basic variable, and introduce the input variable which we think is the innovation point. The classic literature on DEA method and its research mechanism is as follows.

Thirdly, the land fragmentation proposed by reviewer 2 is also within the scope of land, and the labor migration proposed by reviewer 2 also belongs to the category of labor force. In addition, the political vitality and household income sources proposed by reviewer 2 are commonly used in multiple regression measurement methods, so it is not appropriate to introduce them into DEA as variables.

Meanwhile, this paper studies the efficiency of grain production from the perspective of financial support and provincial units. The selected index is agriculture related loans. In previous studies, agriculture related loans can definitely affect grain output. In addition, in the process of grain production in major grain producing areas, funds mainly come from agriculture related loans, which can better explain that agricultural loans will have an impact on this grain production, Therefore, it must be innovative and forward-looking.

(1) Cooper W W, Seiford L M, Tone K. Data envelopment analysis: a comprehensive text with models, applications, references and DEA-Solver software[M]. 2nd ed. New york: Springer Science & Business Media, 2007.

(2) Boussofiane, R.G. Dyson and E. Thanassoulis. Applied data envelopment analysis. European Journal of Operational Research 52 (1991) 1-15.

(3) Kaoru Tone. A slacks-based measure of efficiency in data envelopment analysis. European Journal of Operational Research 130 (2001) 498-509.

(4) Yong-bae Ji, Choonjoo Lee. Data envelopment analysis. The state journal 10 (2010) 267-280.

(5) A. Charnes. W.W. Cooper. Rreface to topic in Data Envelopment Analysis. Annals of Operations Research 2 (1985) 59-94. 

2 The authors seem to mix up the concept of “total factor productivity” with that of “production efficiency”. The former refers to the “overall” contribution of technological progress in the process of economic growth, while the latter refers to the distance of each production unit to the “given” production frontier – they measure different things at different scales. Put differently, even if the production frontier is fixed, you can still estimate production efficiency (for each of the production units), but you cannot estimate total factor productivity in that csae.

The author’s answer: 

Total factor productivity (TFP) is a measure of production efficiency. In this paper, we use the global Malmquist index decomposition method to study TFP. We define total factor productivity as TFP and decompose TFP into technical progress changes (TECH) and efficiency changes (EC). The related literatures are listed below. In order to fully analysis, We further decompose EC into Pure technical efficiency changes (BPC) and scale efficiency changes (SECH). The production efficiency proposed by reviewer 2 should be technical efficiency (EC). We have decomposed EC into SECH and BPC in the manuscript. The formula is as follows:

(1)Pastor JT, Lovell C. A global Malmquist productivity in index. Economics Letters. 2005,88(2):266-271.

(2)Fare R, Grosskopf S, Norris M, Zhang Z(1994). Productivity growth, technical progress, and effciency change in industrialized countries: reply. The American Economic Review. 1997,87(5):1040-1044.

3 The data are not well-suited for the intended analysis. The analysis was based on provincial data. The big question is: Can a province serve as a DMU (decision-making unit) in a production problem? Who is making production-related decisions (involving input choices, management, and even land transfers)? The farmers! Not the provinces. I would say in the 1990s, when farm household data were not readily available, the use of provincial data may be OK; at least, they may reveal some general patterns. However, with tons of farm-level data available in China today, the analysis conducted at the provincial level is largely uninformative, and potentially misleading even. I suspect that the aggregation of input and output data from the farm level to the provincial “kills” a lot of efficiency loss because the averaging exercise tends to drug down the production frontier within a given province.

The author’s answer: 

Reviewer 1 has also raised this question, and we have given a detailed answer. The answers are as follows

We agree with the reviewers that there are indeed restrictions on data selection, using provincial level data does not capture micro- or farm-level variations. However, the samples studied in this paper are 13 major grain producing areas in China, which are divided by provincial units. At the same time, the index data published by the websites and statistical yearbooks of China Bureau of statistics and regional statistical bureaus are not unified, only the provincial data are unified, and there is no unified branch for farm level data. Therefore, in order to ensure the standardization and accuracy of the data, we choose the provincial data to study. We have added it to section "3.2 index selection, sample selection and data sources"

4 The methodology is not sound. There are two major problems. First, the input variables are limited. How about irrigation conditions? How about natural disasters? They all matter for grain production. It’s well-known that the lack of input data will lead to misleading efficiency estimates. This problem is likely to occur in the paper under review.

Second, it’s unclear what role financial support plays in the story. The author treats it as an input in the DEA analysis, but how would loans directly affect agricultural outputs (—I don’t think throwing money in the field would make crops grow)? In theory, it plays the role of releasing the budget constraints of the farmers. Thus, it affects production efficiency mainly through improving “allocative efficiency”, which, however, is not examined in the paper. See, e.g., Smith et al. (2011) for a study examining allocative efficiency.

The author’s answer: 

First of all, due to the limitation of the sample, the reason for the restriction has been explained in detail in the second question raised by reviewer 2. we can't select too many input indicators in the research of DEA method. We also agree with the opinions of the reviewers and add the index of adjustment conditions to the index. As the increase of the index will lead to different results of the model, we have revised the empirical part of the paper. However, we did not accept the natural proposal from the reviewers The first reason is that DEA method limits the number of indicators, adding too many indicators will lead to inaccurate results. The second reason is that there is a certain correlation between natural disasters and irrigation area, and natural disasters do not occur every year. In addition, the impact of irrigation area can be ignored.

Secondly, reviewer 2 thinks that (- I don't think throwing money in the field would make crops grow)? According to the input-output theory based on Walla's general equilibrium theory, in order to balance the relationship between input and output, this manuscript studies the grain production efficiency. The core input indicators are land resources, labor resources and agricultural loans. The output indicator is grain output, which does not involve the issue of resource allocation efficiency. At the same time, we should do research from the specific situation of the field, and serve the practice with theory. In the main grain producing areas, the funds for agricultural production basically come from the agricultural loans of the banks. As long as the farmers in the main grain producing areas apply for loans, the banks will actively examine and approve them. When the farmers get the money, they will carry out grain production and use all the loans for grain production. This process does not involve the allocation Efficiency. The following classical literature on the efficiency of grain production is also based on land resources, labor resources and capital and other indicators, and does not study the allocation efficiency.

(1)Houshyar Ehsan, Azadi Hossein, Almassi Morteza, et al. Sustainable and efficient energy consumption of corn production in Southwest Iran: combination of multi-fuzzy and DEA modeling. Energy. 2012,.44(1):672-681.

(2)Muhammad Arif Watto，Amin Mugera．Measuring efficiency of cotton cultivation in Pakistan: a restricted production frontier study．Journal of the Scienceof Food and Agriculture. 2014,94 (14) : 3038-3045．

(3)Christian Nsiah,Bichaka Fayissa. Trends in Agricultural Production Efficiency and their Implications for Food Security in Sub‐Saharan African Countries. African Development Review. 2019,31(1).

(4) Cooper W W, Seiford L M, Tone K. Data envelopment analysis: a comprehensive text with models, applications, references and DEA-Solver software[M]. 2nd ed. New york: Springer Science & Business Media, 2007.

(5) Boussofiane, R.G. Dyson and E. Thanassoulis. Applied data envelopment analysis. European Journal of Operational Research 52 (1991) 1-15.

(6) Yao, S.J.; Liu, Z.N. Determinants of Grain Production and Technical Efficiency in China. Journal and Agricultural Economics. 1998,49(2):171-184.

5 There are many language problems in the paper, making the paper rather frustrating to read. The authors should check the write-up carefully. Some obvious examples include:

1) “…the previous literature is not only the time span is short” Which is the subject of the sentence, “the literature” or “the time span”?

2) Lines 139-140: “…the innovation points of this paper is mainly in the following three aspects” (problem: Disagreement between the subject and the verb) should be “…the innovation of this paper is mainly in the following three aspects” or “the innovation points of this paper are mainly in the following three aspects”.

The author’s answer: 

First of all, on the whole, we have polished part of the manuscript in the editorial organization cooperating with PLoS One, and revised the language and grammar problems existing in the manuscript. Next, we will answer the language and grammar questions raised by reviewer 2 one by one.

(1)“…the previous literature is not only the time span is short” Which is the subject of the sentence, “the literature” or “the time span”?

The author’s answer: 

The subject of this sentence is a reference in the past. We have changed “…the previous literature is not only the time span is short” to “The time span of previous references is very short”.

(2)Lines 139-140: “…the innovation points of this paper is mainly in the following three aspects” (problem: Disagreement between the subject and the verb) should be “…the innovation of this paper is mainly in the following three aspects” or “the innovation points of this paper are mainly in the following three aspects”.

The author’s answer: 

The reviewer's comments on the amendment are very correct. We have changed “…the innovation points of this paper is mainly in the following three aspects” to “…the innovation of this paper is mainly in the following three aspects”.

We have tried our best to improve the manuscript and made some changes in the manuscript. These changes will not influence the content and framework of the paper. And here we did not list the changes but marked in red in revised paper. We appreciate for Editors and Reviewers’ warm work earnestly, and hope the correction will meet with approval.

Once again, Thank you and the reviewers again for your help.

Yours sincerely,

Sha Lou

---

## [Editor Report · Decision Letter 1]

10 Feb 2021

Research on grain production efficiency  in China ’ s main grain producing areas from the perspective of financial support

PONE-D-20-38013R1

Dear Dr. Sha Lou

We’re pleased to inform you that your manuscript has been judged scientifically suitable for publication and will be formally accepted for publication once it meets all outstanding technical requirements.

Kind regards,

Carlos Alberto Zúniga-González, Ph.D

Academic Editor

PLOS ONE

Additional Editor Comments (optional):

Dear authors, I want to express my sincere congratulations for the effort you have made to improve your manuscript. The issue of productivity and efficiency has been applied very well using the DEA approach with the Malmquist indices. I want to encourage you to continue with this line of research. Therefore my decision is to accept.

Reviewers' comments:

N/A

---

## [Editor Report · Acceptance letter]

2 Mar 2021

PONE-D-20-38013R1 

Research on grain production efficiency in China’s main grain producing areas from the perspective of financial support 

Dear Dr. Lou:

I'm pleased to inform you that your manuscript has been deemed suitable for publication in PLOS ONE. Congratulations! Your manuscript is now with our production department. 

Kind regards, 

on behalf of

Dr. Prof. Carlos Alberto Zúniga-González 

Academic Editor

PLOS ONE